# Payment Behavioral Response Mechanisms for All-Age Retrofitting of Older Communities: A Study among Chinese Residents

**DOI:** 10.3390/bs13110925

**Published:** 2023-11-13

**Authors:** Yang Zhang, Lei Dong

**Affiliations:** College of Management, Xi’an University of Architecture and Technology, Xi’an 710055, China; zhangyang@xauat.edu.cn

**Keywords:** older community renovation, all-age retrofitting, anxiety, PLS-SEM, payment behavioral response mechanisms

## Abstract

Intergenerational integration has given rise to a novel aging paradigm known as all-age communities, which is garnering international attention. In China, the aging population and the implementation of the three-child policy have resulted in increased demand for retirement and childcare services among residents in older neighborhoods. Consequently, there is a pressing need to retrofit these older neighborhoods to accommodate all-age living arrangements given the high demand they generate. Therefore, this study undertakes research interviews with residents and constructs an exploratory theoretical model rooted in established theory. To assess the significance of our model, we employ Smart PLS 3.0 based on 297 empirical data points. Our findings indicate that anxiety has a significant negative effect on payment behavior; objective perception, willingness to pay, and government assistance exert significant positive effects on payment behavior. By comprehensively analyzing the mechanisms underlying residents’ payment behavior, this study provides valuable insights for the government for promoting the aging process within communities and formulating effective transformation policies.

## 1. Introduction

The all-age community concept, originally coined as “lifetime homes” by the Joseph Rowntree Foundation and Habinteg Housing Association in the 1990s [1], has gained prominence in addressing the evolving needs of older localities in China. It means that the community should provide residents of all ages with living space, service facilities, and comfortable public spaces, and create a healthy residential area where residents have equal opportunities to participate in daily decision-making. In August 2021, the Chinese Ministry of Housing and Urban-Rural Development (MOHURD) issued a document to strictly regulate large-scale demolition and construction activities during the urban renewal process. Consequently, there is a growing requirement for all-age retrofitting to meet the practical and policy requirements of older localities. At present, China’s aging population situation is grim. According to the forecast of the Chinese Academy of Social Sciences, China will become the country with the highest degree of population aging in the world by 2030 [2]. The recent implementation of the three-child birth policy has led to an increase in the number of children, while the dilapidated infrastructure and single-purpose facilities in older communities pose challenges in meeting the material and psychological needs of the older adults and children under these new circumstances [3]. Given the aging population and the three-child policy, an all-age retrofitting approach focused on intergenerational integration aligns more closely with public demand. Previous studies on all-age community transformation have focused on constructing child- and age-friendly cities, emphasizing developmental goals and facilities [4]. At the mesoscopic scale, this study proposes recommendations for land-use structure, area index, and facility configuration based on the daily activities of the older population and their satisfaction with the environment, adopting the perspectives of living circles and communities [5]. This project aimed to provide essential facilities and services [6], well-organized transportation systems [7], and functional layouts [8] to the older population. The design principles and recommendations are rooted in the life circle and community perspectives. At the micro level, specific design considerations, such as pedestrian space [9], housing planning and design [10], and accessibility [11], received particular attention. The ultimate goal is to construct an all-age community that fosters positive intergenerational interaction, providing residents of all ages with improved living, working, and growing opportunities that enhance their social interactions [12].

A comprehensive literature review of the current research indicated a predominant focus on retrofitting programs, policy recommendations, and visionary goals. However, there is a noticeable dearth of studies examining on residents’ willingness to participate in retrofitting initiatives. Previous studies on willingness to retrofit have identified influential factors by synthesizing the available literature [13]. Nevertheless, these studies have not fully addressed all the influencing factors that may exist beyond the scope of prior research. To address this gap, the present study employs a resident-oriented approach by conducting interviews and drawing on relevant theoretical frameworks to identify influencing factors. These factors serve as the foundation for constructing a theoretical model that analyzes the mechanisms underlying residents’ payment behavior. Presently, the Chinese government assumes the financial burden associated with renovating old communities; however, given the considerable scale of these renovations, solely relying on government financing poses limitations and hampers the progress of these endeavors. The analysis of its internal mechanism has practical significance and can provide a reference for the government to guide the residents of old communities to actively cooperate with the transformation work, promote the process of community aging, and formulate community transformation policies.

## 2. Methodology and Data Sources

The existing literature indicates the absence of established measurement scales, variable categories, and theoretical hypotheses for studying payment response mechanisms in the context of all-age retrofitting [14]. Empirical research conducted in the field has revealed that residents in older neighborhoods hold diverse understandings of all-age retrofitting, and quantitative research results obtained from large-scale resident surveys based on undifferentiated structured questionnaires may lack validity in capturing their perspectives effectively [15]. To address these limitations, this study designed an unstructured (open-ended) questionnaire to conduct interviews with a representative sample of residents from older neighborhoods, thereby collecting primary information [16]. A qualitative approach was employed to explore the payment response mechanism more effectively [17]. Figure 1 shows the research framework of payment response mechanisms.

The selection of specific interviewees was guided by a theoretical sampling approach aligned with the theoretical development of the research design [18]. To capture the genuine perspectives from residents, a random selection process was used to identify 24 participants aged 20–60, with financial capability, as interviewees from five representative old neighborhoods in Xi’an and Chengdu, China. Following the concept of information saturation, no new coded content emerged after the inclusion of 24 interviewees, and 2 additional interviewees were recruited to ensure data saturation. Therefore, the final sample size was 24, ensuring a reasonable level of reliability in the obtained results [19]. Detailed information regarding the interviewed participants is provided in Table 1.

In-depth personal interviews, each lasting approximately 30 min, were conducted as part of the study. The purpose of these interviews was to gain a comprehensive understanding of the interviewees’ attitudes, emotions, and underlying motivations regarding the funding of all-age transformation. During the individualized interviews, the interviewer carefully observed the interviewee’s external expressions and internal psychological states, aiming to create an environment that allowed ample space for reflection and expression [20]. Prior consent was obtained from the interviewees for audio recording. Following the completion of the interviews, the recorded material was meticulously collated, transcribed, and thoroughly reviewed, resulting in a substantial collection of interview transcripts exceeding 10,000 words. Two-thirds of the interview transcripts (16) were randomly selected for coding analysis and model construction, while the remaining transcripts (8) were reserved for testing theoretical saturation.

The present study employed a qualitative research design and adopted an exploratory approach rooted in theory to examine the response mechanism of residents in old neighborhoods towards all-age retrofitting payments. The construction of a theoretical model involved three distinct stages: open, spindle, and selective coding [21], which were facilitated by the use of NVivo 11 software. A continuous comparative analysis was adopted to analyze the data, following the guidelines outlined by the authors of reference [22]. Furthermore, NVivo 11 was used to manage and analyze the qualitative data obtained from the in-depth interviews, aiming to identify the primary factors influencing residents’ payment behavior.

### 2.1. Category Refinement

#### 2.1.1. Release Codes

Open coding, also known as level 1 coding, involves systematically tagging, coding, and meticulously documenting the initial interview transcripts verbatim, aiming to generate the initial concept from the raw material [23]. Throughout the coding process, careful consideration was given to mitigate personal biases and prejudices by extracting initial concepts as labels directly from the interviewees’ original statements. A comprehensive collection of over 103 original statements and their corresponding initial concepts were obtained through primary extraction. Given the large number of initial concepts and their repetitive nature, a recategorization and regrouping exercise was undertaken [24]. Only the initial concepts that recurred three or more times were retained, while inconsistent or less frequently occurring initial concepts were excluded. For brevity, Table 2 provides a summary of the retained initial concepts and their corresponding categories. To conserve space, three representative information statements and their corresponding initial concepts were selected for each category.

#### 2.1.2. Spindle Coding

Main axis coding, also referred to as associative registration, was employed to explore the inherent logical connections between the initial categories, thereby expanding the main categories and their respective subcategories [25]. By systematically examining the interrelationships and logical sequence of the categories at the conceptual level [26], we organized them into distinct groups. As a result of this coding process, four main categories were identified, each encompassing a range of open coding categories. The comprehensive details of this classification, including the main categories and their associated open coding categories, are presented in Table 3.

#### 2.1.3. Selective Coding

Selective coding, referred to as core logging, was employed to identify and elucidate the core categories derived from the main categories. The core categories were analyzed in relation to the main categories as well as other relevant categories. The purpose was to depict the underlying phenomena and behavioral conditions in a coherent and narrative manner, resulting in the development of a novel substantive theoretical framework [27,28].

This study identified the core category of “payment response mechanisms for all-age adaptations,” which served as the central focus for the subsequent narrative. The resulting “storyline” can be summarized as follows: the three main categories, namely anxiety, objective perception, and external situation, all of which exert a significant influence on the payment response mechanisms. Anxiety emerged as an internal driving force that inversely impacts residents’ willingness to pay for all-age adaptations, whereas objective perceptions and external situation act as moderating factors, shaping individuals’ willingness to pay. Drawing upon this “storyline”, we constructed and developed a model of payment response mechanism known as the “All-age Payment Response Mechanism Model” or the “Situation-Willingness-Response Integration Model” (SWRM). This model captures the interplay and integration of various factors within the payment response mechanism. The intricate relationship structures of the main categories in this study are presented in detail in Table 4.

#### 2.1.4. Mediating Variables

Ajzen [29] suggests that individual behavioral intentions are not always seamlessly translated into behavioral responses, indicating that external factors, such as facilitative conditions, also play a significant role in this process [30].

Drawing upon the theory of interpersonal behavior, the present study introduces the mediating variable “government assistance” within the theoretical model, aiming to account for the mediating effect it exerts between willingness to pay and the actual payment response. Figure 2 illustrates the proposed theoretical model.

### 2.2. Willingness to Pay Influencing Factors Analysis

Table 5 presents detailed descriptions of the ten primary influencing factors, which were extracted using the previous category extraction process.

### 2.3. Structural Equation Modeling Analysis

#### 2.3.1. Research Hypothesis

To analyze the structure of the constructed payment response mechanism, a structural equation model was selected as the analytical framework to assess the direct and indirect effects of relevant factors [41]. This selection was motivated by the capability of structural equations to integrate several similar potential variables into a unified “block variable” and then analyze their causal relationships. Consequently, this method aligns with the analytical rationale of this study and offers an ideal approach to analyze multiple influencing factors.

Through a thorough theoretical analysis, this study determined that the factors influencing residents’ all-age modification payment response could be summarized into three main categories: anxiety, objective perception, and external situation. To ensure the scientific validity and rationality of the theoretical model, an empirical research method utilizing structural equation modeling was employed. This involved formulating research hypotheses, designing questionnaires, and collecting and analyzing data to validate the model. Building upon the previously constructed all-age payment response mechanism model, five variables were identified—objective perception (OP), anxiety (AN), external situation (EC), willingness to pay (WTP), and government support (GS)—and eight research hypotheses were proposed. (Figure 3).

**H1.** 
*The objective perceptions of residents regarding community all-age transformation have a significant positive effect on their willingness to pay.*


**H2.** 
*The objective perceptions of residents regarding community all-age transformation have a significant positive effect on their payment response.*


**H3.** 
*The objective perceptions of residents regarding community all-age transformation have a significant inverse effect on their anxiety.*


**H4.** 
*The anxiety experienced by residents in community all-age transformation has a significant inverse effect on their willingness to pay.*


**H5.** 
*The external situation in which residents are placed during community all-age transformation has a significant inverse effect on their anxiety.*


**H6.** 
*The external situation in which residents are placed during community all-age transformation has a significant positive effect on their willingness to pay.*


**H7.** 
*The willingness of residents to pay in community all-age transformation has a significant positive effect on their behavioral response.*


**H8.** 
*Government support plays a positive mediating role in the relationship between residents’ government intentions to payment response.*


#### 2.3.2. Questionnaire Design and Data Collection

The questionnaire was designed using 10 latent variables, employing a scale based on theoretical categories. To ensure optimal reliability and validity, all indicators were carefully selected from established scales, with reactive and formative indicators revised for the two cost study components. Reactive indicators, such as group pressure, transformational trust, and government assistance, were utilized, while formative indicators, including perceived value, performance, economic cost, and risk, were incorporated. The Likert 7 subscale [42] was adopted, with response ratings ranging from 1 to 7, corresponding to “completely disagree” to “completely agree”. Higher scores indicated a greater level of agreement (Table 6).

To enhance data collection efficiency and the validity of the collected data, control measures were implemented in three key aspects: district sampling, targeted population selection for questionnaire distribution, and careful monitoring of the filling process. The MOHURD has initiated pilot urban regeneration projects in 21 cities (districts) that are highly representative and exemplify these pilot areas. Stratified sampling was employed [50], with the country divided into four regions: East, Central, West, and Northeast. A random sample of approximately 10% was drawn from each region, encompassing 21 cities, including Beijing, Nanjing, Xi’an, and Chengdu. This regional-based sampling approach helps overcome the issue of geographical concentration in sampling, resulting in improved sample representativeness, statistical data reliability, and enhanced external validity.

The questionnaires were distributed among middle-aged and older individuals residing in the neighborhood who required both aging care and childcare and possessed the financial capacity to pay. Prior to conducting the survey, participants were provided with a comprehensive briefing on the research objectives, questionnaire content, and instructions on how to effectively complete the questionnaire. Emphasis was placed on the individualized completion of the questionnaire to ensure authentic responses and minimize potential biases, such as social approval and expectancy effects, which could influence the questionnaire data.

A total of 340 questionnaires were distributed, resulting in the collection of 329 completed questionnaires through the employment of the instant return method, yielding a commendable return rate of 96.75%. Subsequently, after excluding 18 questionnaires containing incomplete responses resulting from objective factors, such as insufficient information pertaining to renovation, as well as 14 invalid questionnaires with noticeable patterns or errors, a final sample of 297 valid questionnaires remained, accounting for 90.27% of the initial distribution. The demographic characteristics of the respondents are presented in Table 7.

#### 2.3.3. Measurement Model Testing

Reliability and validity were initially assessed in this study using different methods to test formative and reactive indicators. Reactive indicators involved a causal relationship between the measured items, with each item expected to exhibit internal consistency, interchangeability, and moderate to high correlation [51]. Conversely, formative indicators establish a causal relationship between item and indicator, meaning that removing any item can alter the potential index, and item correlations may become negative [52]. The reliability of reactivity indicators was empirically assessed through combined reliability (CR) and Cronbach’s coefficient (Cronbach’s alpha). Generally, values above 0.7 for Cronbach’s alpha and CR indicate high reliability and convergence effects. These criteria are accompanied by a standardized factor loading estimate (Std.) greater than 0.7 and an average variance extracted (AVE) above 0.5. Discriminant validity measurement requires that the AVE for each potential indicator surpass the square of the correlation (e) (Table 8). Similarly, differential validity measurement requires that the AVE for each potential indicator exceed the square of the correlation of each potential indicator (Table 9). To establish the validity of formative indicators, their reliability indicators are typically examined. Specifically, item weights should exceed 0.2, demonstrate significance, and exhibit t-values greater than 1.96. Additionally, because formative indicators exhibit negative regression, covariance should be assessed, ensuring a VIF below 5.

#### 2.3.4. Structural Model Testing

The parameters were estimated using partial least squares, and the structural model was tested using Smart PLS 3.0 to perform path analysis [53]. The final model’s factor path diagram is depicted in Figure 4. The results of hypothesis testing are presented in Table 10. The model successfully passed the significance test at or below the 5% level, indicating the reliability of the theoretical model (Figure 4). In the context of all-age retrofitting in old neighborhoods, which encompasses anxiety, objective perceptions, and external situation, the payment response is influenced by the mediating variable of willingness to pay. Notably, anxiety exhibits the most substantial effect on willingness to pay. On the one hand, willingness to pay directly translates into payment response, while on the other hand, it indirectly affects payment response through government assistance. Moreover, residents’ objective perceptions significantly and positively impact behavioral responses. Therefore, residents’ payment behavior in the all-age transformation of old neighborhoods can be characterized as “spontaneous” (willingness to pay → payment response), “induced” (willingness to pay → government support → payment response), and “constrained” (objective perception → payment response), with the dominant behavioral logic being “spontaneous”.

## 3. Conclusions

The interview text of 24 residents of old communities was deeply coded, and the influencing factors of residents’ payment response were refined, and a theoretical model was constructed based on this basis. The variables of willingness to pay were anxiety, objective perception, and external situation, and the variables of payment response were willingness to pay, objective perception, and government.

The empirical testing of the “Situation-Willingness-Response” model was conducted to explore its applicability to the payment response mechanisms of community residents. The findings demonstrated that objective perception, anxiety, and external situation exert a significant influence on residents’ willingness to pay. Notably, anxiety exhibited the most substantial effect, indicating that higher levels of anxiety are associated with decreased willingness to pay. Furthermore, objective perceptions, willingness to pay, and government assistance were found to have a significant positive impact on payment response. The dominant factor shaping residents’ payment behavior was their willingness to pay, while government support and objective perceptions displayed a similar effect.

The factor path diagram of the model demonstrates the significant positive correlation between objective perception and willingness to pay, indicating that an increase in perceived value and economic cost corresponds to a higher willingness to pay among individuals. Moreover, the inclusion of government assistance significantly improves payment behavior, suggesting that, in Chinese societies characterized by high levels of public trust, citizens’ behavioral decisions are primarily influenced by their organization-oriented perceptions of trust in government assistance rather than issue-oriented perceptions of event risk. The reliability of government assistance plays a crucial role in strengthening residents’ willingness to pay.

## 4. Discussion

### 4.1. Payment Behavioral Study

A comprehensive literature review of the current research shows that current research on the renovation of old residential areas focuses on renovation projects, policy recommendations, and visionary goals. However, there is a significant lack of research on residents’ willingness to participate in renovation programs and their payment behavior. The paper makes a substantial contribution to academic discourse by revealing the multifaceted determinants that influence the payment behavior of residents within the field of all-age renovation. By classifying key variables and building a robust theoretical model, the manuscript improves our understanding of how residents respond to the needs of community transformation. This knowledge has the potential to inform decision-making processes related to urban renewal and inspire further academic exploration in this area.

### 4.2. Recommendations

To elucidate the intricate relationship among multiple factors influencing residents and their payment response, this study develops a comprehensive model of the factors that shape payment response, drawing upon foundational theoretical frameworks. Specifically, the model incorporates anxiety as a psychological response indicator and government support as a component of interpersonal behavior theory, thereby enriching our understanding of residents’ payment response mechanisms in the context of all-age transformation. Additionally, it highlights the imperative of providing support during the all-age retrofitting process in older neighborhoods, aiming to improve residents’ payment response in the following aspects:

(1) Strengthen residents’ perception of the renovation value, alleviate their anxiety surrounding the renovation process, and foster a more spontaneous payment response. Residents should be effectively informed regarding the direct benefits they can derive from the renovations and emphasize the close link between renovation funding and the effectiveness of the renovation efforts. Simultaneously, the government should proactively employ methods such as household surveys, community visits, and internet opinion analysis to meticulously scrutinize residents’ discontent and grievances. Moreover, residents should be made aware of the pressing need to address the challenges posed by an aging population and the implementation of policies, such as the three-child policy, for family retirement and childcare. Specifically, it is essential to convey to residents that the deteriorating conditions and impractical layout planning of old neighborhoods are unsuitable for fulfilling the needs of aging care and childcare. Furthermore, the positive impact of all-age retrofitting on neighborhoods’ suitability for such purposes, along with highlighting how the implementation of renovations can alleviate anxiety, will be vital in garnering residents’ support and cooperation.

(2) Strengthen government support to improve the “inducibility” of payment response in all-age retrofitting. Given the positive effect of government assistance on payment response, it is recommended that the government streamline the ownership adjustment procedures, actively engage residents in the retrofitting process, facilitate knowledge sharing regarding neighborhood retrofitting accomplishments, and disseminate information about the retrofitting program. By prioritizing such measures, the government can effectively reduce the perceived difficulty in payment response, diminish anticipated risks, and provide residents with the necessary support and convenience to actively participate in the retrofitting process.

(3) Improving residents’ capacity for action in retrofitting endeavors and alleviating the “constraint” on their payment response. In the early stages of all-age retrofitting in older neighborhoods, the government can actively engage residents by mobilizing their enthusiasm and motivation to participate in the retrofitting process. A participatory approach can be adopted, whereby residents are invited to contribute their preferences and choices regarding the desired improvements. Moreover, collaboration between the government and neighborhood committees can establish a monitoring group responsible for overseeing renovation proposals. This inclusive approach empowers residents to deeply involve themselves in formulating renovation plans in the early stages, exercising control over the progress of the retrofitting process, and managing and maintaining renovated areas in the later stages. By fostering such resident involvement, the government facilitates greater ownership and agency among residents, thereby facilitating a more effective and less constrained payment response.

(4) Clearly delineate the distinctions between different age groups. Aging care focuses on the process of providing care and support for elderly individuals, aiming to promote their well-being and recovery. On the other hand, childcare is centered around nurturing and raising children with the objective of ensuring their successful development. Understanding and acknowledging the divergences between the two groups in terms of required efforts, service orientations, and willingness for development is crucial. By recognizing these differences and proposing appropriate solutions, it becomes possible to improve residents’ payment behavior.

### 4.3. Limitations and Future Directions

Despite the valuable insights gained from this study, it is important to acknowledge certain limitations that should be considered for future research. The study’s findings may be subject to biases because the interviewees and questionnaire respondents were mainly from older neighborhoods in China, which introduces the possibility of biased findings due to the specific national context. To address this limitation, future research should include residents from older neighborhoods in different countries to analyze the payment response mechanisms within their respective national contexts. Second, the analysis of the effects of demographic variables on the hypotheses presented in the paper did not discuss the impact of how demographic factors such as education level, age, and gender intersect with residents’ payment behavior and influence payment behavior in the context of full age modification. Additionally, collecting longitudinal data at multiple time points would allow for the exploration of the dynamic interaction between emotions and cognition as well as its impact on residents’ behavior. Lastly, it is important to note that this study focused on residents’ intentions rather than their actual payment behavior. As intentions and behaviors may differ [54], further research could examine the relationship between residents’ payment intentions and their actual payment behavior.

## Figures and Tables

**Figure 1 behavsci-13-00925-f001:**
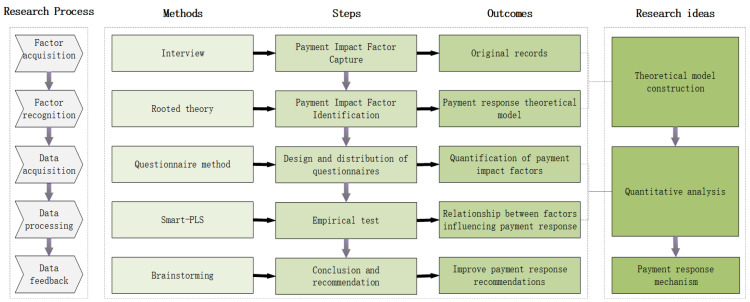
Research framework of payment response mechanisms.

**Figure 2 behavsci-13-00925-f002:**
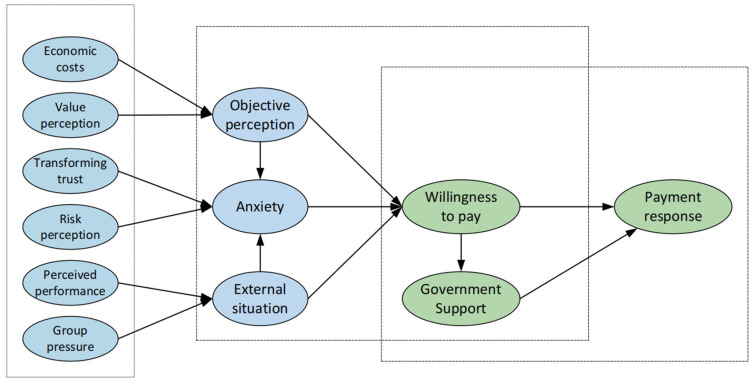
Situation-willingness-response integration model.

**Figure 3 behavsci-13-00925-f003:**
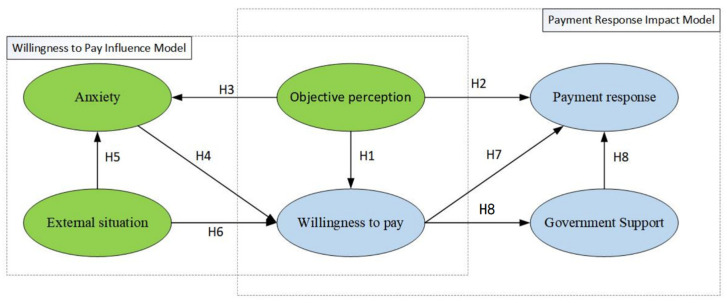
Hypothetical model of all-age payment response mechanism.

**Figure 4 behavsci-13-00925-f004:**
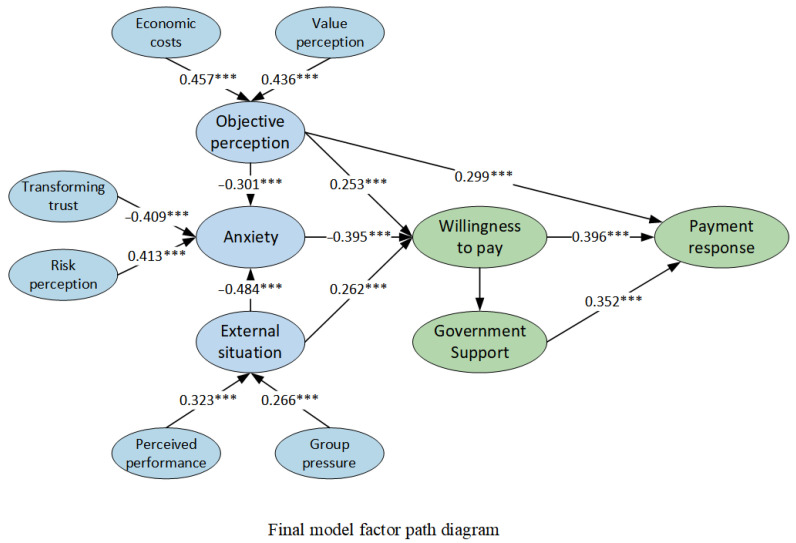
Final model factor path diagram. Note: “***” indicate significance at the 0.1% levels, respectively.

**Table 1 behavsci-13-00925-t001:** List of basic information about the interviewees.

Respondent Serial Number	Respondents	Gender	Age	Academic Qualifications	Occupation
Respondent 01	Mr. Tang	Male	25	Masters	Postgraduate student
Respondent 02	Mr. An	Male	62	Specialties	Fitter
Respondent 03	Ms. Wang	Female	72	Junior High School	Factory worker
Respondent 04	Mr. Liu	Male	54	Specialties	Electrical sales
Respondent 05	Mr. Lee	Male	62	Specialties	Materials management
Respondent 06	Mr. Yang	Male	31	Undergraduate	Antiques
Respondent 07	Ms. Zhang	Female	44	Specialties	Design department
Respondent 08	Mr. Lui	Male	48	Specialties	Factory worker
Respondent 09	Mr. Zhao	Male	44	Specialties	Factory worker
Respondent 10	Mr. Wang	Male	39	Undergraduate	Finance
Respondent 11	Ms. Chan	Female	31	Undergraduate	Factory worker
Respondent 12	Mr. Zhao	Male	24	Undergraduate	Construction industry
Respondent 13	Mr. Zhang	Male	26	Undergraduate	Financial sector
Respondent 14	Ms. Wang	Female	23	Undergraduate	Construction industry
Respondent 15	Ms. Zhang	Female	23	Specialties	Sales
Respondent 16	Mr. Wang	Male	33	Undergraduate	Foreign trade
Respondent 17	Mr. Liu	Male	25	Masters	Postgraduate student
Respondent 18	Mr. Wang	Male	23	Undergraduate	Estimator
Respondent 19	Mr. Locke	Male	24	Masters	Postgraduate student
Respondent 20	Ms. Lee	Female	25	Masters	Physical distribution management
Respondent 21	Mr. Lee	Male	32	Undergraduate	Project Manager
Respondent 22	Mr. Choi	Male	33	PhD	University teacher
Respondent 23	Miss Chow	Female	28	Undergraduate	Accounting
Respondent 24	Miss Shang	Female	21	Undergraduate	Financial practitioner

**Table 2 behavsci-13-00925-t002:** Open coding scoping.

Category	Original Statement (Initial Concept)
Risk perception (RP)	A01 How much do you have to charge for which services after the renovation (financial risk) A09 What to do if your family doesn’t adapt (adaptation risks) A01 Where this money is spent, we don’t know (risk of benefit)
Transforming trust (TT)	A01 It is generally considered that it will be more convenient to wait for your family to get old (remodeling brings convenience) A03 I am definitely willing to pay for the renovation if it is done well (the renovation is good) A13 feels pretty good if it’s really like you say it is (the makeover fits the need)
Economic costs (EC)	A01 If you want to pay for it, just change the plumbing and electricity or something (to cover the cost of the basic type of renovation) A09 It’s too much effort to go up and down the stairs in this old neighborhood, I think we can install a lift and spend money on it (to pay for the cost of renovation of the perfect class) A18 I have no one to take care of the children at home now, if I can have childcare I can pay appropriately (to cover the cost of upgrading the class)
Value perception (VP)	A07 I’m getting older now, I live on the fifth floor and I’d still like to have a lift added to make it easier (more convenient to get around) A15 If the community can be renovated, the children can have a better play environment, the elderly can take a walk or something, now there is nothing in the community (renovation to provide convenience) A16 to maintain the infrastructure, for example, like the water supply, and then the electricity, that is, at the very least not to have power and water cuts (to improve the situation)
Perceived performance (PP)	A03 I don’t know if this modification has ever been successful (case impact) A07 I am willing to fund this as long as the scheme is good. (Actual transformation effect) A09 will also have noise, negative impact of housing devaluation (tangential benefits)
Group pressure (GP)	A01 must consider what other people think, after all, we all live together (the influence of other people’s wishes) A09 If everyone around us agrees to the change, then we have to listen to the majority opinion (the influence of others’ wishes) A19 The wishes of my surrounding neighbors are still important to me (the influence of the wishes of others)

**Table 3 behavsci-13-00925-t003:** Master categories formed using spindle coding.

Main Category	Corresponding Categories	The Connotation of Relationships
Anxiety (AN)	Risk perception	Residents’ perception of the risks of retrofitting can induce anxiety, reducing their willingness to pay for retrofitting
	Transforming trust	Residents’ trust in all-age rehabilitation programs reduces their own anxiety and influences their psychological sense of support for rehabilitation
Objective perception (OP)	Economic costs	The financial cost to residents of participating in the renovation (incurred by the various renovation projects) will affect their willingness to pay for the renovation
	Value perception	The value (use, economic, emotional value) that the renovation generates for the neighborhood affects the willingness of residents to pay for the renovation
External Situation (EC)	Perceived performance	The impact of completed retrofit cases on residents’ willingness to pay for retrofitting (i.e., whether the results meet residents’ expectations) can affect residents’ willingness to pay for retrofitting
	Group pressure	The influence of social atmosphere and the willingness of others on residents’ willingness to pay for renovations

**Table 4 behavsci-13-00925-t004:** Typical relational structure of the main categories.

Typical Relationship Structure	Relationship Structure Connotation
Anxiety→Willingness	Anxiety is an endogenous trigger for residents to pay for the response and can have an internal inducing effect on whether they pay for the retrofit
Objective Perception→Willingness	Impact of perception of objective environment (cost of participation, improvement in living experience by renovation) on willingness to pay
External Situation→Willingness	The influence of perceptions of external situations (transformation effects, group pressure) on willingness to pay

**Table 5 behavsci-13-00925-t005:** Explanatory table of influencing factors.

Influencing Factors	Theoretical Definitions	Operational Definitions
Risk perception (RP)	Slovic defines risk perception as “the subjective assessment of future losses and uncertainties formed by residents based on their personal experiences and attributes” [31].	In the context, this refers to residents’ perceptions of the cost, effectiveness, and ability to adapt to the lifestyle risks of retrofitting activities.
Transforming trust (TT)	In Trust and Power, the sociologist Luhmann argues that trust is an expectation generalized by relying on information beyond what is available, and that “in its broadest sense, trust refers to confidence in someone’s expectations, and it is a fundamental fact of social life” [32].	Drawing on Rousseau and others, this paper defines trust in government as “the belief in the reliability and dependability of government departments and the expectation and belief that government will protect the public interest in the face of uncertainty”.
Economic costs (EC)	Costs are the monetary representation and objectification of resources that must be expended in order to carry out a production activity or to achieve a certain purpose. In another sense, cost can also be the price that must be paid for a choice (Ref. [33]).	Participation costs include the economic costs for residents to participate in the different types of renovation (basic category improvement category upgrading category).
Value perception (VP)	The concept of perceived value was first introduced by Zeithaml in 1988 in the theory of perceived value from the customer’s perspective, where she defined perceived value as the customer’s overall assessment of the utility of a product or service when the perceived benefit is weighed against the cost of acquiring the product or service (Ref. [34]). She defines perceived value as the customer’s overall evaluation of the usefulness of a product or service when the benefits are weighed against the cost of obtaining it.	In this paper, we refer to the extent to which residents feel that making all-age modifications would improve the current situation of care for the elderly and for children.
Perceived performance (PP)	Perceived performance refers to the level of performance that consumers can perceive as meeting their needs when experiencing a product or service [35].	In the text, this refers to the impact of residents’ perceptions of the effectiveness of retrofitting on their payment response.
Group pressure (GP)	Group pressure is the perceived psychological oppression that occurs when an individual’s intentions conflict with the norms of the group. The group in which an individual lives is one of the most important considerations when making a purchase decision, and the group can modify the individual’s behavior through the exchange of information and the guidance of values, which can have a model and constraining effect on the individual’s choices (Ref. [36]).	In the context, it refers to the influence of the wishes of neighbors, family, and friends around the resident on the resident’s response.
Anxiety (AN)	“Anxiety state”, which refers to the degree of anxiety experienced by an individual at a given moment in a given situation. “Anxiety quality”, which refers to the psychological characteristics of an individual’s personality in terms of anxiety disposition, i.e., an aspect of personality [37].	The text refers to the negative emotions of residents facing the risks of renovation and the uncertainty caused by trust issues.
Objective perception (OP)	The series of processes by which the conscious mind perceives, senses, and notices information about the objective world. Perception can be divided into sensory processes and perceptual processes [38].	The text refers to residents’ perceptions of the cost of participating in the transformation and the value generated.
External Situation (EC)	External Situations include macro and micro factors, with macro contexts generally referring to external political, social, technological, and economic contexts [39].	The text abstracts group pressure, government support, and perceived performance as External Situations. They give residents influence from the outside.
Government Support (GS)	Government support is the conscious activity of the government in regulating economic and social life by various means in order to achieve the desired objectives [40].	The text refers to the impact of government policies and subsidies on rehabilitation on the response of residents.

**Table 6 behavsci-13-00925-t006:** Questionnaire on payment response mechanisms for residents in older neighborhoods.

Potential Variables	Serial Number	Title Item	Reference Sources
Risk perception	RP1	I am concerned about the increase in expenditure after the renovation	[43]
	RP2	I am concerned about not adapting to the modified lifestyle (adaptation risk)	
	RP3	I am concerned that the use of the fee is not disclosed and transparent (risk of benefit)	
Transforming trust	TT1	I believe that rehabilitation can solve problems and conflicts	[43]
	TT2	I believe the renovation program and process is sound	
	TT3	I believe that when an all-age transformation is carried out it is for the good of all	
Economic costs	EC1	I am willing to pay for the economic costs of infrastructure improvements (water, electricity, gas, heating, etc.)	[44]
	EC2	I am willing to pay the economic costs of the improvement category (additional barrier-free access, additional lifts, additional charging posts, greening of the plot, etc.)	
	EC3	I am willing to pay the economic costs of upgrading the type of renovation (elderly care, health services, convenient markets, etc.)	
Value perception	VP1	The renovated community is more suitable for retirement and childcare than it is now (use value)	Research
	VP2	House prices will rise after the conversion (economic value)	
	VP3	Closer to family after remodeling (emotional value)	
Perceived performance	PP1	The inconvenience of living in my current neighborhood makes me willing to fund the renovation	Research
	PP2	Knowledge of completed renovation cases will influence my thoughts	
	PP3	Comments from residents of the completed renovation block can influence my thoughts	
Group pressure	GP1	Family and friends support the renovation and I will consider funding it	[45]
	GP2	I will consider funding the renovation with vigorous publicity from the residents’ committee and the media	
	GP3	I would consider contributing if the majority of my neighbors were willing to do so	
Government support	GS1	I would consider funding if the government provided policy support	[46]
	GS2	I would consider contributing if the government provided financial support	
	GS3	I would consider funding a renovation if the government made the process of monitoring it public	
Anxiety	AN1	Worried and apprehensive after learning of the renovation	[47]
	AN2	Nervous and unsettled to learn of the renovation	
	AN3	The remodel has to deal with a lot of things and feel annoyed	
Objective perception	OP1	The renovated neighborhood makes me feel happy physically and mentally	[48]
	OP2	Acceptable inconveniences during renovation	
	OP3	Adequate funding allows for better results and a faster transformation process	
External situation	ES1	The completed renovation of the community has been very effective	[48]
	ES2	The government is actively promoting the renovation to proceed	
	ES3	Growing support for renovation	
Willingness to pay	WTP1	I am willing to pay for the renovation	[49]
	WTP2	I will pay for the renovation	
	WTP3	I will do my best to cover the cost of the renovation	
Payment response	PR	Level of positive payment	

**Table 7 behavsci-13-00925-t007:** Demographic data of respondents (N = 297).

Demographic Variables	Number	Percentage
Gender		
Male	130	43.77%
Female	167	56.23%
Age		
Under 30 years old	86	36.14%
30–55 years	123	37.39%
55+ years	120	36.47%
Region		
East	132	44.44%
Western	23	7.74%
Central	112	37.71%
North East	30	10.10%
Level of education		
Undergraduate	36	12.12%
Undergraduate	228	76.77%
Postgraduate and above	33	11.11%
Annual household income (USD 10,000)		
Below 5	16	5.39%
5–15	101	34.01%
15–30	133	44.78%
Over 30	47	15.82%

**Table 8 behavsci-13-00925-t008:** Index factor loading and average extraction variance.

	Cronbach’s Alpha	rhoA	Composite Reliability	Average Variance Extracted (AVE)
VP	N.A.	1.000	N.A.	N.A.
EC	0.777	0.781	0.870	0.691
OP	0.761	0.761	0.863	0.678
PP	N.A.	1.000	N.A.	N.A.
PR	1.000	1.000	1.000	1.000
WTP	0.857	0.859	0.913	0.777
TT	0.847	0.855	0.907	0.765
GS	0.788	0.805	0.876	0.702
AN	0.862	0.864	0.916	0.784
EC	N.A.	1.000	N.A.	N.A.
GP	0.743	0.808	0.847	0.649
RP	N.A.	1.000	N.A.	N.A.

Note: N.A. means not applicable.

**Table 9 behavsci-13-00925-t009:** Differential validity of measurement models.

	VP	EC	OP	PP	RP	WTP	TT	GS	AN	EC	GP	RP
VP	N.A.											
EC	0.301	0.831										
OP	0.436	0.472	0.823									
PP	0.196	0.323	0.171	N.A.								
PR	0.453	0.547	0.666	0.272	1.000							
WTP	0.380	0.565	0.656	0.249	0.791	0.882						
TT	0.359	0.306	0.195	0.162	0.393	0.360	0.875					
GS	0.374	0.409	0.429	0.213	0.672	0.549	0.463	0.838				
AN	−0.213	−0.626	−0.530	−0.126	−0.594	−0.614	−0.410	−0.443	0.886			
EC	0.374	0.414	0.457	0.409	0.480	0.468	0.383	0.373	−0.347	N.A.		
GP	0.361	0.267	0.232	0.050	0.294	0.219	0.318	0.209	−0.121	0.362	0.805	
RP	−0.038	−0.310	−0.296	−0.083	−0.365	−0.333	−0.207	−0.259	0.414	−0.332	−0.210	N.A.

Note: N.A. means not applicable.

**Table 10 behavsci-13-00925-t010:** Hypothesis test results.

Research Hypothesis	Estimate	*t*-Value	P	Conclusion
H1	0.253	7.201	***	Support
H2	0.299	6.006	***	Support
H3	−0.301	5.878	***	Support
H4	−0.465	4.404	***	Support
H5	−0.484	10.569	***	Support
H6	0.200	3.693	***	Support
H7	0.475	8.655	***	Support
H8	0.324	6.590	***	Support

Note: “***” indicate significance at the 0.1% levels, respectively.

## Data Availability

The data presented in this study are available in Appendix A.

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
