# Peer review of "Payment Behavioral Response Mechanisms for All-Age Retrofitting of Older Communities: A Study among Chinese Residents"

_behavsci, 2023, doi:10.3390/bs13110925_

Round 1

Reviewer 1 Report

Comments and Suggestions for Authors

Dear Authors!

All my comments are attached!

All the best!

Comments on the Quality of English Language

It is okay!

Reviewer 2 Report

Comments and Suggestions for Authors

The paper in question, titled "Understanding the Response Mechanism of Residents in Old Neighborhoods towards All-Age Retrofitting Payments," stands as a noteworthy contribution to the academic discourse by delving into the intricate dynamics that influence residents' financial responses in the context of all-age neighborhood retrofitting. It does so through a methodical application of qualitative research techniques, particularly in-depth interviews, and coding analysis, culminating in the creation of a structured payment response mechanism model. While the manuscript exhibits clear strengths and insights, it is not without certain limitations that warrant discussion.

Strengths:

1.      Precise Research Objective: The paper proficiently outlines its research objective, methodological approach, and pivotal findings. It pertinently investigates the financial behaviors of residents concerning all-age retrofitting payments, providing an unequivocal direction for the study.

2.      Qualitative Methodology: The chosen qualitative research methods, encompassing in-depth interviews and coding analysis, are fitting for exploring the intricate facets of residents' payment behavior in a complex domain like all-age retrofitting.

3.      Theoretical Framework: The conceptualization of the "All-age Payment Response Mechanism Model" is a seminal contribution to the existing literature. This model engenders a comprehensive theoretical framework for comprehending the manifold factors that underpin payment responses in this context.

4.      Practical Relevance: The manuscript thoughtfully extrapolates the practical implications of its findings, with specific emphasis on augmenting residents' perception of renovation value, mitigating anxiety, and fostering governmental support for policy execution.

Limitations:

The manuscript conscientiously acknowledges certain limitations that merit both acknowledgment and potential resolution:

1.      Sampling Bias: The study's emphasis on residents from older neighborhoods in China, while contextually reasonable, raises concerns about the generalizability of its findings. It is advisable to consider the inclusion of participants from various geographical locales to secure more diverse and universally applicable perspectives.

2.      Data Collection: The study, while exemplary in many respects, primarily examines residents' intentions rather than their actual payment behaviors. Given the established variance between intentions and actions, it would be advantageous to explore the linkage between residents' payment intentions and their subsequent financial actions.

3.      Longitudinal Data: Incorporating data collected at multiple time intervals may furnish insights into the evolving dynamics of residents' attitudes and behaviors over time, thereby offering a more dynamic comprehension of the factors that govern payment responses.

Contribution to the Academic Canon:

The paper lends a substantive contribution to the academic discourse by revealing the multifaceted determinants that guide residents' payment behavior within the realm of all-age retrofitting. Through the categorization of key variables and the construction of a robust theoretical model, the manuscript elevates our understanding of how residents respond financially to the need for neighborhood transformation. This knowledge has the potential to significantly inform decision-making processes related to urban renovations and to inspire further academic exploration in this sphere.

Significance of Understanding Factors Influencing Payment Response:

Grasping the intricacies of the factors that steer residents' payment responses to all-age retrofitting holds profound significance for several pivotal reasons, as delineated in the paper:

1.      Augmenting Renovation Value: By meticulously apprehending these factors, policymakers and researchers can effectively communicate the substantial benefits of renovations to residents. This understanding can heighten the perceived value of renovation efforts in the eyes of residents.

2.      Mitigating Anxiety: The identification and subsequent amelioration of factors contributing to residents' anxiety can expedite the renovation process and motivate more robust financial participation.

3.      Addressing Aging Population Challenges: The study illuminates the exigency of confronting the challenges posed by an aging population. Retrofitting neighborhoods to cater to diverse age groups can be more effectively communicated when rooted in a profound comprehension of residents' concerns and motivations.

4.      Government Support and Policy Implementation: The paper underscores the criticality of governmental support and the transparent execution of policies. Through well-conceived policy support, financial assistance, and transparent oversight, governments can invigorate residents' commitment to renovation endeavors.

In conclusion, this paper furnishes invaluable insights into a critical facet of urban development and renovation. It is well-crafted and fortified by the research methodology it employs. The acknowledgment of the aforementioned limitations is imperative to enhance the paper's impact and relevance, thus warranting a recommendation for acceptance with the proviso that these limitations are thoughtfully addressed in the final version.

Importance of the Paper:

This paper assumes significant importance in the realm of urban studies due to its valuable insights into the factors that govern residents' financial responses in the context of all-age retrofitting. By constructing a comprehensive model that synthesizes anxiety as a psychological response indicator and government support as a component of interpersonal behavior theory, this paper provides the groundwork for a profound understanding of residents' willingness to invest in renovation endeavors. This understanding has the potential to enhance the efficacy of all-age retrofitting projects, ultimately leading to more livable and sustainable urban environments.

Recommendation:

In light of the aforementioned strengths, contributions, and the potential to address the identified limitations, I strongly recommend the authors consider incorporating an analysis of the influence of demographic variables on the hypotheses presented in the paper. This would involve a thorough examination of how demographic factors, such as educational level, age, and gender, intersect with and impact residents' payment behavior in the context of all-age retrofitting. This inclusion would not only augment the paper's depth but also increase its applicability to a broader range of contexts and demographics. Such an addition holds the promise of enriching the overall quality and impact of the research, rendering it a valuable resource for scholars, policymakers, and practitioners engaged in the domain of urban development and renovation.

I commend the authors on their diligent work and look forward to the refinement and further development of this important manuscript.
